# Rapid Response to Experimental Warming of a Microbial Community Inhabiting High Arctic Patterned Ground Soil

**DOI:** 10.3390/biology11121819

**Published:** 2022-12-14

**Authors:** Kevin K. Newsham, Birgitte Kortegaard Danielsen, Elisabeth Machteld Biersma, Bo Elberling, Guy Hillyard, Priyanka Kumari, Anders Priemé, Cheolwoon Woo, Naomichi Yamamoto

**Affiliations:** 1British Antarctic Survey, NERC, High Cross, Madingley Road, Cambridge CB3 0ET, UK; 2Center for Permafrost, Department of Geosciences and Natural Resource Management, University of Copenhagen, Øster Volgade 10, DK-1350 Copenhagen, Denmark; 3Department of Environmental Health Sciences, Graduate School of Public Health, Seoul National University, Gwanak-ro, Gwanak-gu, Seoul 08826, Republic of Korea; 4Department of Biology, University of Copenhagen, Universitetsparken 15, DK-2100 Copenhagen, Denmark

**Keywords:** carbon dioxide (CO_2_), climate change, cryoturbation, frost boils, greenhouse gases, methane (CH_4_), non-sorted circles, soil bacteria, soil fungi, Svalbard

## Abstract

**Simple Summary:**

Surface temperatures in the Arctic are rising more rapidly than elsewhere on Earth. Precipitation patterns in the region are also altering, with an increased incidence of heavy rainfall. However, the effects of warming and increased water availability on soil microbes—which have pivotal roles in nutrient cycling and greenhouse gas exchange—inhabiting the sparsely vegetated patterned ground soils that are widespread across the High Arctic are not well understood. Here, we warmed patterned ground soils on Svalbard with open top chambers and irrigated the soils twice each summer. After four years, a 1 °C rise in summertime near-surface soil temperature affected the exchange of the greenhouse gases carbon dioxide (CO_2_) and methane (CH_4_), with warmed soils emitting 44% more CO_2_, and consuming 78% more CH_4_, than soils that were not warmed. Warming also increased soil bacterial abundance by 32%, and, of the 40 most abundant bacterial taxa, led to both reductions and increases in the relative abundances of four taxa. Irrigation did not influence the measured variables. At the current rate of summertime warming in soils on Svalbard (0.8 °C per decade), we anticipate that these effects will become apparent in the natural environment by approximately the mid 2030s.

**Abstract:**

The influence of climate change on microbial communities inhabiting the sparsely vegetated patterned ground soils that are widespread across the High Arctic is poorly understood. Here, in a four-year experiment on Svalbard, we warmed patterned ground soil with open top chambers and biannually irrigated the soil to predict the responses of its microbial community to rising temperatures and precipitation. A 1 °C rise in summertime soil temperature caused 44% and 78% increases in CO_2_ efflux and CH_4_ consumption, respectively, and a 32% increase in the frequency of bacterial 16S ribosomal RNA genes. Bacterial alpha diversity was unaffected by the treatments, but, of the 40 most frequent bacterial taxa, warming caused 44–45% reductions in the relative abundances of a *Sphingomonas* sp. and *Ferruginibacter* sp. and 33–91% increases in those of a *Phenylobacterium* sp. and a member of the *Acetobacteraceae*. Warming did not influence the frequency of fungal internal transcribed spacer 2 copies, and irrigation had no effects on the measured variables. Our study suggests rapid changes to the activities and abundances of microbes, and particularly bacteria, in High Arctic patterned ground soils as they warm. At current rates of soil warming on Svalbard (0.8 °C per decade), we anticipate that similar effects to those reported here will manifest themselves in the natural environment by approximately the mid 2030s.

## 1. Introduction

Polar amplification has caused mean annual surface air temperatures in the Arctic to increase at a faster rate than the global average over the last 50 years [1,2,3]. On the High Arctic archipelago of Svalbard, mean annual surface air temperatures have risen at 0.7 °C per decade over the last half century [4], and since the 1990s have increased at 1.7 °C per decade, approximating to twice the rate of temperature change for the Arctic and seven times that of the global average over the same period [5]. Mean annual soil temperatures on Svalbard have similarly risen by 3.6 °C between 1998 and 2017 [6]. Because the potential amount of precipitable water in the atmosphere is associated with temperature, precipitation events in the Arctic are now also more frequent than in previous decades [3], with rain set to become the dominant form of precipitation in the region by the end of the 21st Century [7]. Accordingly, 3–8% increases in total annual precipitation received per decade have been recorded at several meteorological stations on Svalbard during the last century, with future projections based on RCP8.5 simulations indicating an increase in the heaviest rainfall events on the archipelago during summer and autumn [4,8].

Soil bacteria and fungi have pivotal roles in terrestrial ecosystem processes, such as the cycling of nutrients and the exchange of greenhouse gases with the atmosphere [9]. Determining how warming and increased precipitation will influence soil microbes is hence fundamental to understanding how Arctic terrestrial ecosystems will alter in the future [10]. However, although much information has accumulated in the literature on microbial community responses to experimental warming and irrigation in densely vegetated Arctic soils, e.g., [11,12,13,14,15,16,17], little is known of how the activities and abundances of the microbes inhabiting sparsely vegetated patterned ground soils in the Arctic will be affected by climate change. Patterned ground features, consisting of frost boils (also known as non-sorted circles), hummocks and polygons, account for significant areas of land in the region [18], where they have key roles in nutrient turnover, soil development and active layer dynamics [19,20,21]. The cryoturbation that produces these features restricts plant cover, with vegetation usually being absent from the centres of High Arctic frost boils and polygons [21]. However, current knowledge of how microbial communities in these predominantly barren soils will respond to warming and precipitation is limited, since the few previous studies that have examined the effects of increased temperature and irrigation on microbial activity in patterned ground soils have been confined to those with relatively high (*c*. 50%) plant cover [22,23].

Here, we report the results of a four-year field experiment designed to predict how the activities and abundances of microbes inhabiting a sparsely vegetated patterned ground soil on Svalbard will respond to experimental warming and irrigation. We anticipated that, as in vegetated Arctic soils, warming and irrigation would rapidly affect the rates of exchange of greenhouse gases between the atmosphere and soil [23,24]. However, in contrast with the majority of studies on densely vegetated Arctic soils [11,12,13,14,15,16,17], but in accordance with those on barren soils in maritime Antarctica [25,26,27], we anticipated rapid treatment effects on the size and composition of the soil microbial community.

## 2. Materials and Methods

### 2.1. Site Description and Field Experiment

The experiment was set up in September 2014 at Kongsfjordneset (78°57.997′ N, 11°28.545′ E) on the Brøgger Peninsula, Svalbard (Figure 1a). Average summer, winter and annual air temperatures on the peninsula are 3.7 °C, −12.7 °C and −5.7 °C, respectively, and mean annual and summertime precipitation are currently *c*. 550 mm and 82 mm, respectively [4]. Snow cover accumulates from mid September [28] and at Kongsfjordneset has usually melted by early June. The bedrock at Kongsfjordneset is carbonate, with dolomite being the main mineral present [29]. The landscape is characterised by frequent patterned ground with frost boils [19] and is a cryptogam herb barren with very sparse, low-growing plant cover (unit B1; [18]). Biological soil crusts and crustose lichens (typically *Ochrolechia frigida* (Sw.) Lynge) sparsely colonise the soil in frost boils, and vascular plants, typically *Salix polaris* Wahlenb. but also *Bistorta vivipara* (L.) Delarbe and *Saxifraga oppositifolia* L., grow at the peripheries of the boils. Soil collected in September 2014 from frost boils at 0–30 mm depth had a pH value of 8.06 (±0.02), carbon (C) and nitrogen (N) concentrations of 8.28% (±0.08) and 0.15% (±0.01), respectively, and a C:N ratio of 57.04 (±2.21) [30].

The field experiment consists of 48 plots centred on individual frost boils (Appendix A) treated with a factorial combination of warming with open top chambers (OTCs; Figure 1b) and irrigation. The experimental design results in four OTC-irrigation treatments, each replicated 12 times across three blocks [31]. The irrigation treatment consisted of applying 1 L of deionised water to 24 of the frost boils in mid–late June and late August each year, simulating *c*. 20 mm rainfall events. Prior to the experiments reported here, the most recent irrigation treatment had been applied on 4 July 2018.

### 2.2. Long-Term Soil Temperature Measurements

Temperatures were monitored by Tinytag Plus 2 loggers (TGP-4017, Gemini Data Loggers Ltd., Chichester, UK) installed in soil in four chambered and four unchambered plots. The loggers recorded temperatures at a depth of 30–35 mm between 10 September 2014 and 27 August 2018. They were replaced yearly with newly calibrated units. Measurements recorded between September 2017 and August 2018 by two loggers that had become exposed at the soil surface in summer 2018 were deleted from the dataset.

### 2.3. Gas Exchange and Instantaneous Soil Temperature and Soil Water Content Measurements

Gas exchange between the atmosphere and soil was measured twice in each of the 48 frost boils, on 23 and 26 August 2018, using a closed loop system and a Piccaro Gas Analyzer (Picarro G4301, Santa Clara, CA, USA). The analyzer was attached to a transparent polycarbonate chamber equipped with fans for air circulation seated on stainless steel frames that had been hammered into the soil in each boil on 30 June 2018 (Figure 1b and Appendix A). Gutters filled with water around each frame ensured an airtight seal (Appendix A). Analyses of images (ImageJ, US National Institutes of Health, Bethesda, MD, USA) taken in August 2018 indicated that the mean (±SEM) cover by lichens and biological soil crusts within the frames was 3.97% (±0.53), and that by vascular plants was 0.46% (±0.21). CO_2_ and CH_4_ exchange was measured over a period of 5 min, during which the chamber was covered with a dark cloth to eliminate photosynthetically active radiation. CO_2_ effluxes thus represent ecosystem respiration [23]. The fluxes of the gases were calculated by fitting 2nd order polynomial models to changes in gas concentrations over time. In order to avoid bias associated with the initial stabilization period and saturation towards the end of the measurements, only data measured from 50–250 s were included in these calculations, and gas fluxes were calculated from slopes taken 100 s after the start of each measurement [32]. Soil temperature and volumetric soil water content (VSWC) were measured at three points in each frost boil at 2 cm and 5 cm depths during the measurements using a soil thermometer (Hanna Instruments, Leighton Buzzard, UK) and a Theta probe (ML3, Delta-T Devices Ltd., Cambridge, UK), respectively. In order to avoid pseudoreplication [33], gas exchange data were averaged over the two samplings, and the temperature and VSWC measurements were averaged over the three points, two depths and two samplings prior to inclusion in further analyses.

### 2.4. Soil Sampling

Following the gas exchange measurements on 26 August 2018, barren soils (0–30 mm depth) were sampled from several places in each frost boil using sterile spoons, were bulked into a sterile Petri dish and were frozen at −20 °C within 3 h of sampling, prior to the analyses described below.

### 2.5. DNA Extractions

DNA was extracted from defrosted sub-samples of soil (1.1 g fresh weight) under a sterile hood using a PowerSoil DNA isolation kit (MoBio Laboratories, Carlsbad, CA, USA). Eluents were split into two aliquots, which were subsequently dried under a sterile hood for 48 h in round-bottomed sterile 96 well plates.

### 2.6. Q-PCR Assays

The copy number of bacterial 16S ribosomal RNA genes in rehydrated aliquots was measured in 20 µL reactions, consisting of 0.8 µL of each of the primers 341F (5′-CCTAYGGGRBGCASCAG-3′) and 806R (5′-GGACTACHVGGGTWTCTAAT-3′), 10 µL of 2 × qPCRBIO SyGreen Blue Mix Lo-ROX (PCR Biosystems Inc., Wayne, PA, USA), 2 µL of sample (diluted 10 times to avoid inhibition of PCR) and 6.4 µL of H_2_O. The PCR mixes were heated to 95 °C for 180 s, and then subjected to 45 cycles of 95 °C for 5 s and a final melt at 60 °C for 30 s on a LightCycler^®^ 96 real-time PCR instrument (Roche Life Science, Hvidovre, Denmark). Fungal ITS2 copy numbers were measured in the same way, but with 0.8 µL of each of the primers ITS4 (5′-TTCCTSCGCTTATTGATATGC-3′) and ITS7 (5′-GTGARTCATCGARTCTTTG-3′) in 20 µL reactions. The measurements from one sample, for which the copy numbers of bacterial 16S ribosomal RNA genes and fungal ITS2 regions were 2–3 orders of magnitude lower than the other 47 samples, were deleted from the dataset. Copy numbers were expressed per g dry weight (dwt) soil (105 °C for 18 h).

### 2.7. Barcoding of Bacterial 16S Ribosomal RNA Genes

DNA in aliquots was resuspended in 50 µL of Tris–EDTA buffer. A universal eubacterial primer set, 340F (5′-TCCTACGGGAGGCAGCAGT-3′) and 800R (5′-GGACTACCAGGGTATCTAATCCTGTT-3′) [34], was used to amplify V3–V4 hypervariable regions of the 16S ribosomal RNA gene with MiSeq adapters. Each PCR was carried out in a 50 μL reaction volume containing 2 × Premix Taq™ (Takara Bio Inc., Otsu, Shiga, Japan), 1 μM of each primer and 1 μL of DNA extract. PCR amplification was performed in a BioRad T100™ thermal cycler (Bio-Rad Laboratories, Inc., Hercules, CA, USA) with an initial denaturation step at 95 °C for 5 min, followed by 35 cycles of denaturation for 15 s at 95 °C, annealing for 45 s at 56 °C and elongation for 90 s at 72 °C, and a final elongation step for 10 min at 72 °C. The PCR products were purified using AMPure XP beads (Beckman Coulter, Inc., Brea, CA, USA) and a second PCR step was performed to ligate unique dual-index adapters with each sample using a Nextera XT Index Kit v2 (Illumina, Inc., San Diego, CA, USA). The second PCR step was performed in a 25 μL reaction volume containing 2 × Premix Taq™ (Takara Bio Inc.), 5 μL of each index primer and 5 μL of each purified PCR product. The thermal cycling conditions were set to 95 °C for 3 min followed by eight cycles of 95 °C for 30 s, 55 °C for 30 s and 72 °C for 30 s, with a final elongation step at 72 °C for 5 min. The final libraries were purified using AMPure XP beads (Beckman Coulter) in 30 μL of 10 mM Tris–HCl (pH 8.5), and were pooled in equimolar concentrations (4 nM) before sequencing on an Illumina MiSeq sequencer (2 × 300 bp) (Illumina, Inc.). DNA was successfully amplified from 43 samples, which, in total, generated 2,459,819 paired sequences.

The demultiplexed raw sequence reads were trimmed with Trimmomatic version 0.35 [35] using the settings SLIDINGWINDOW:4:5, MINLEN:36, and subsequently analyzed following the MiSeq SOP in mothur v1.40.5 [36]. The trimmed paired ends sequence reads were merged and a non-redundant collection of sequences was generated by binning identical sequences. The resulting unique sequences were aligned against a SILVA-based reference alignment [37] and sequences differing by up to two basepairs were preclustered [38]. Chimeric sequences were detected using VSEARCH implementation in mothur and removed [39]. The taxonomy of the high-quality 16S ribosomal RNA gene sequences was assigned against the EzBiocloud database [40] using naïve Bayesian classification [41]. Sequences were clustered into OTUs at a 97% similarity cutoff using the OptiClust implementation in mothur [42]. Singleton OTUs were omitted and the resulting operational taxonomic unit (OTU) table was subsampled randomly at 12,463 sequences per sample prior to alpha diversity calculations.

### 2.8. Statistical Analyses

Data were tested for normality using Kolmogorov–Smirnov tests. Mean annual and seasonal temperatures measured in chambered and unchambered soils were compared using Student’s *t*-tests. Other variables were tested for the main effects of OTCs, irrigation and block and the OTC × irrigation interaction using general linear models (GLMs). When necessary, data were normalised using log_10_ (*x* + 1)-transformations prior to analysis, or, if they failed normality tests following transformation, were analysed using GLMs with bootstrapping (10,000 randomizations). GLMs for gas exchange included VSWC as a continuous covariate. Permutational multivariate analysis of variance (PERMANOVA) and environmental fitting analyses on Hellinger-transformed Bray–Curtis distances were used to determine associations between bacterial community composition and treatments, block and VSWC. Analyses on the relative abundances of bacterial OTUs were restricted to the 40 most abundant taxa, and were corrected for multiple comparisons using the Benjamini-Hochberg procedure [43]. Pearson’s correlations were used to determine associations between variables. Statistical analyses were performed using MINITAB 19 (State College, PA, USA) and SPSS version 28.0.0.0 (IBM).

## 3. Results

### 3.1. Soil Temperatures

Long-term measurements indicated that soils at Kongsfjordneset were frozen for 63% of the year, with their mean temperature remaining permanently below 0 °C for 6.5–7.5 months, typically from mid–late October until late May–early June (Figure 2a). They were permanently thawed for approximately three months each year, usually between June and late August–early September (Figure 2a). Minimum and maximum temperatures recorded in unchambered soils were −17.2 °C and 18.4 °C, and those in chambered soils were −14.0 °C and 20.8 °C, respectively (Figure 2a). The OTCs increased annual soil temperature by 0.6 °C relative to unchambered soil (T value 4.90, *p* = 0.004; Figure 2b). No effects of OTCs were recorded on soil temperature in spring (MAM; *p* = 0.112, T value = −1.93; Figure 2b), with the majority of the warming occurring when soils were thawed in summer (JJA), when a 1.0 °C increase in soil temperature was recorded in OTCs (T value = 7.86, *p* = 0.001; Figure 2b). Increases in soil temperature of 0.4 °C were also recorded in chambers during autumn (SON; T value 3.47, *p* = 0.018), but OTCs only had a marginally significant effect on soil temperature in winter (DJF; *p* = 0.069, T value = −2.78; Figure 2b). The OTCs also increased soil temperature during the samplings in August 2018, with mean temperatures measured in unchambered and chambered soils of 6.1 °C and 7.3 °C, respectively (*F*_1,42_ = 210.9, *p* < 0.001).

### 3.2. Gas Fluxes and Associations with Soil Temperature and VSWC

The OTCs increased CO_2_ efflux from soil. Compared with unchambered soils, the chambers elicited a 44% increase in mean CO_2_ release, with mean effluxes of 0.081 and 0.117 μmol CO_2_ m^−2^ s^−1^ from unchambered and chambered soils, respectively (*F*_1,41_ = 25.66, *p* < 0.001; Figure 3a). There was also a marginally significant positive effect of irrigation on CO_2_ efflux (*F*_1,41_ = 3.31, *p* = 0.076), with mean efflux values of 0.092 and 0.106 μmol CO_2_ m^−2^ s^−1^ being recorded from unirrigated and irrigated soils, respectively. The efflux of CO_2_ was unaffected by the OTC × irrigation interaction (*F*_1,41_ = 0.02, *p* = 0.881; Figure 3a). Correlative analyses indicated a positive association between CO_2_ efflux and soil temperature during the samplings (slope = 0.021 μmol CO_2_ m^−2^ s^−1^, *r* = 0.507, *p* < 0.001; Figure 3a, inset). GLMs with bootstrapping also indicated that CH_4_ uptake into soil was affected by OTCs, with a 78% increase in the mean rate of CH_4_ consumption by chambered soils compared with unchambered soils (mean values −1.62 and −0.91 μmol CH_4_ m^−2^ h^−1^, respectively; *F*_1,41_ = 4.083, *p* = 0.050; Figure 3b). The rate of CH_4_ consumption was unaffected by irrigation or the OTC × irrigation interaction (both *F*_1,41_ ≤ 0.410, *p* ≥ 0.526; Figure 3b) and was only marginally significantly correlated with mean soil temperature measured during the samplings (*r* = −0.262, *p* = 0.072; Appendix A). Mean VSWC measured during the samplings was 0.28 m^3^ water m^−3^ soil (Appendix A) and was unaffected by OTCs, irrigation or the OTC × irrigation interaction (all *F*_1,42_ ≤ 2.09, *p* ≥ 0.155). VSWC was not a significant covariate in GLMs (both *F*_1,41_ ≤ 2.83, *p* ≥ 0.100) and correlative analyses similarly showed no associations between VSWC and CO_2_ efflux or CH_4_ uptake (both *r* ≤ 0.178, *p* ≥ 0.204).

### 3.3. Q-PCR Assays

After accounting for a significant block effect (*F*_2,41_ = 34.87, *p* < 0.001), OTCs were found to increase the number of 16S ribosomal RNA gene copies g^−1^ dwt soil, with a 32% rise in the number of 16S copies in chambered soil compared with unchambered soil (mean values 3.50 × 10^8^ and 2.65 × 10^8^ copies g^−1^ dwt soil, respectively; *F*_1,41_ = 4.35, *p* = 0.043; Figure 4). No effects of irrigation or the OTC × irrigation interaction were recorded on the number of 16S gene copies g^−1^ dwt soil (both *F*_1,41_ ≤ 0.56, *p* ≥ 0.460; Figure 4). GLMs with bootstrapping showed that the number of ITS2 copies g^−1^ dwt soil, whilst exhibiting similar responses to the number of 16S gene copies in soil, were not significantly affected by OTCs (*F*_1,41_ = 2.81, *p* = 0.101; Appendix A). Irrigation and the OTC × irrigation interaction also did not influence ITS2 copy numbers g^−1^ dwt soil (both *F*_1,41_ ≤ 0.20, *p* ≥ 0.660, Appendix A).

### 3.4. Soil Bacterial Community Composition and Alpha Diversity

At the class level, the soil bacterial community was dominated by the α-Proteobacteria (Appendix A), which accounted for 35.8% of the community. The β-Proteobacteria, γ-Proteobacteria, Acidimicrobiia, Actinobacteria, Blastocatellia, Gemmatimonadetes, Nitriliruptoria, Rubrobacteria, Solibacteres, Spartobacteria, Sphingobacteriia and Thermoleophilia each accounted for 2.3–8.8% of the community and, along with the α-Proteobacteria, the 40 most abundant OTUs recorded in soil (Appendix A; Figure 5). Archaea were not detected, but type II methanotrophs, such as *Methylocystis* (*Methylocystaceae*) and members of the *Beijerinckiaceae*, were recorded in soil. Mean bacterial OTU richness across all treatments was 2323 (range 1975–2592; Appendix A). No main or interaction effects of OTCs and irrigation were recorded on OTU richness, or on the Shannon, inverse Simpson or Chao1 diversity indices (all *F*_1,37_ ≤ 2.61, *p* ≥ 0.115; Appendix A). PERMANOVA and environmental fitting analyses of Bray–Curtis distances similarly indicated no significant effects of OTCs, irrigation, block or VSWC on soil bacterial community composition (Appendix A; Appendix A).

### 3.5. Individual Soil Bacterial Taxa

The 40 most frequent bacterial taxa responded more strongly to the OTC treatment than to irrigation (Figure 5). Following Benjamini-Hochberg correction, analyses using GLMs with bootstrapping indicated that the relative abundances of four bacterial taxa responded significantly to OTCs and that none of the taxa responded to irrigation (Figure 5). The relative abundance of a member of the genus *Sphingomonas* (OTU 7, α-Proteobacteria) was reduced by 45% in soil sampled from OTCs, from a mean of 1.55% in unchambered soil to 0.86% in chambered soil (*F*_1,37_ = 13.31, *p* = 0.0008; Figure 6a). A member of *Ferruginibacter* (OTU 40, Thermoleophilia) was similarly found to be 44% less abundant in chambered soil than in unchambered soil, in which its mean relative abundances were 0.45% and 0.25%, respectively (*F*_1,37_ = 9.96, *p* = 0.0034; Figure 6b). In contrast, the relative abundances of two OTUs were increased by the OTC treatment. The relative abundance of a member of the family *Acetobacteraceae* (OTU 26, α-Proteobacteria) increased by 91%, from a mean of 0.33% in unchambered soil to 0.63% in chambered soil (*F*_1,37_ = 13.13, *p* = 0.0009; Figure 6c). That of a member of the genus *Phenylobacterium* (OTU 34, α-Proteobacteria) also increased by 33%, from a mean of 0.33% in chambered soil to 0.44% in unchambered soil (*F*_1,37_ = 9.66, *p* = 0.0043; Figure 6d). An OTC × irrigation interaction effect, caused by the greater increase in the abundance of OTU 26 in response to OTCs in unirrigated than in irrigated soils, was also recorded (*F*_1,37_ = 4.36, *p* = 0.043; Figure 6c).

## 4. Discussion

In agreement with previous studies on Arctic soils [23,44,45,46], we found that a 1 °C rise in summertime near-surface temperature in a sparsely vegetated patterned ground soil on Svalbard increased soil CO_2_ efflux and CH_4_ consumption rates after 4 years of treatment. However, in contrast with experiments on vegetated soils in the Arctic [11,12,13,14,15,16,17], but in accordance with those on barren maritime Antarctic soils [25,26,27], rapid impacts of the warming treatment on the size and composition of the soil microbial community were recorded. Other than a marginally significant positive effect on CO_2_ efflux, irrigation had no effects on greenhouse gas flux rates or the soil microbial community. These observations are discussed below in the light of the rapid warming to which soils on the Brøgger Peninsula are currently exposed [4,6].

### 4.1. Increased CO_2_ Efflux from Warmed High Arctic Patterned Ground Soil

Previous studies have shown increased effluxes of CO_2_ from vegetated Low- and High Arctic soils warmed with OTCs [23,45,46], most probably indicating faster rates of soil organic C decomposition by microbes. Similarly, we found that OTCs, which had no measurable effects on VSWC but increased soil temperature by 1.0 °C during summer and by 1.2 °C during the samplings, led to greater CO_2_ efflux from soil, with, relative to unchambered soils, an increase of 0.036 μmol CO_2_ m^−2^ s^−1^ emitted from warmed soil. A linear increase in CO_2_ efflux of 0.021 μmol m^−2^ s^−1^ per degree Celcius rise in temperature was also measured during sampling. Despite the highly significant increase in CO_2_ efflux rate from warmed soil, the measured effluxes from unchambered and chambered soils (0.081 and 0.117 μmol CO_2_ m^−2^ s^−1^, respectively) were lower than those from other vegetated High Arctic soils (0.2–3.0 μmol CO_2_ m^−2^ s^−1^; [47,48,49]). Given the limited soil organic C stocks on the Brøgger Peninsula, which are estimated at just 1.3 kg C m^−2^ in the upper 0.1 m of soil in deglaciated areas [19], and the low estimate of total permafrost C stock on Svalbard (0.11 Pg C; [50]), it seems unlikely that patterned ground soils with sparse plant cover will become substantial C sources as temperatures rise on the archipelago. The expansions in plant communities that are expected on the peninsula as it warms, which will increase overall CO_2_ sink strength [51,52], will further limit C emissions from its terrestrial habitats. In addition, accelerated cryoturbation in patterned ground soils as they warm may redistribute surface organic C to the subsoil, where it will become less biologically available, mitigating the loss of CO_2_ to the atmosphere from increased soil respiration [53].

### 4.2. Increased CH_4_ Consumption by Warmed High Arctic Patterned Ground Soil

We recorded a 78% increase in CH_4_ consumption by warmed soils, with mean uptake rates into unchambered and chambered soils of −0.91 and −1.62 μmol CH_4_ m^−2^ h^−1^, respectively. This observation supports the mounting evidence that dry and mesic High Arctic soils with VSWCs of 0.07–0.32 m^3^ water m^−3^ will act not as sources of CH_4_ as they warm, but as sinks for the gas [44,54,55]. However, the rates of CH_4_ consumption by soil at Kongsfjordneset are low compared with six other locations in the Arctic, where mean flux rates of between −0.9 and −8.9 μmol CH_4_ m^−2^ h^−1^ have been measured [55], suggesting that dry or mesic soils on the Brøgger Peninsula are unlikely to act as substantial sinks for the gas as they warm. The reason for increased CH_4_ consumption by warmed soil is presently unclear, but may arise from relative changes to the activities of soil methanogenic Archaea or methanotrophic bacteria, which evolve and oxidise the gas, respectively. It seems unlikely that the increased uptake of CH_4_ by warmed soil is caused by decreased CH_4_ evolution by Archaea, which inhabit wet, anaerobic soils at depths of >100 mm on the peninsula [56], since OTCs have diminishing effects on temperatures at greater depths in the soil profile [57]. Increased activities of aerobic type II methanotrophs, such as *Methylocystis* spp., which were recorded here and are present in other soils at ≤100 mm depth on the Brøgger Peninsula [58], instead offer a more plausible explanation for the increased consumption of CH_4_ by warmed soil at Kongsfjordneset. However, since previous studies have shown the responses of soil CH_4_ consumption rates to temperature to be relatively small (Q_10_ values of 1.1–1.4 in temperate forest soils and 1.7–2.1 in Arctic tundra or barren soils; [54,59,60,61]), the increase in CH_4_ uptake by warmed soil is unlikely to have arisen from the physiological responses of methanotrophic bacteria to increased temperature alone.

### 4.3. Soil Microbial Community Responses to Warming

Although meta-analyses indicate positive effects of warming on soil microbial abundance in cold regions [62,63], empirical studies that have passively warmed vegetated Arctic soils during summer for up to 25 years have typically recorded few, or sparse, effects of increased temperature on microbial community size (e.g., [11,14,15,16,17]). Here, we recorded effects of OTCs on bacterial community size in a patterned ground soil within 4 years of treatment. At present, the reasons for the differences in the responses to warming of the soil microbial communities in the present and previous empirical studies are not clear. However, the rapid response of the bacterial community to warming in the sparsely vegetated soil studied here is similar to that of microbial communities in barren Antarctic fellfield soils warmed with OTCs, which respond to treatment within 5 years [25,26,27]. The relatively rapid response of the microbial community to warming reported here corroborates the view that delays of up to 15 years in microbial responses to elevated temperature in vegetated Arctic soils arise from gradual changes to the biomass and composition of the plant community [11]. However, the low concentration of N relative to C in the soil at Kongsfjordneset may also explain the responsiveness of its bacterial community to warming. A recent study found that warming increases 16S ribosomal RNA gene copy number within 7 years of treatment in soils with initial C:N ratios of >37 [17]. The observations here, from a soil with an initial C:N ratio of 57, thus corroborate this finding. We advocate further research to identify whether strong inhibitory effects on decomposition processes and the subsequent immobilisation of N in microbial protein [9] might help to explain the increase in bacterial abundance in warmed soils with high C:N ratios. Further studies should also determine why snowfences, which increase wintertime soil temperatures, also have rapid (1–6 years) effects on microbial communities in vegetated Arctic soils [64,65,66].

### 4.4. Responses of Individual Soil Bacterial Taxa to Warming

Despite the harsh environmental conditions on the Brøgger Peninsula, a relatively high bacterial OTU richness of 2323 was recorded in the alkaline (pH 8) soil at Kongsfjordneset, which is consistent with soil bacterial alpha diversity peaking in neutral to mildly alkaline soils [67]. In agreement with previous experiments that have used OTCs to warm Arctic soils for 8–16 years [13,68], we recorded no effects of warming on the alpha diversity of the soil bacterial community after 4 years of treatment. However, despite a uniform soil bacterial community composition across the 48 plots at the start of the experiment in 2014 [31], the relative abundances of four of the 40 most frequent OTUs differed between chambered and unchambered soils by 2018, suggesting subtle, but measurable, changes to the soil bacterial community elicited by warming. The relative abundances of two taxa, a *Sphingomonas* sp. and a *Ferruginibacter* sp., were lower in warmed soil. Given the ability of these genera to degrade polyaromatic hydrocarbons and chitin [69,70], their decreased abundances are suggestive of reduced turnover of complex organic compounds in warmed soil. Another two taxa, a member of the *Acetobacteraceae* and a *Phenylobacterium* sp., were more abundant in warmed soils. Increased abundances of *Acetobacteraceae* in chambered soil at Kongsfjordneset indicate possible beneficial effects of warming on the synthesis of organic acids and the fixation of N_2_ from the atmosphere [71]. However, the markedly limited nutritional spectrum of members of the genus *Phenylobacterium*, which are unable to utilise most sugars, alcohols, amino or carboxylic acids [72], suggests that increases in their abundance may not be beneficial to the turnover of C and N in warmer High Arctic soils.

### 4.5. Delayed Response of the Soil Fungal Community to Warming

In contrast to the effects recorded on soil bacterial community size, the analyses here showed no significant impacts of warming or irrigation on soil fungal ITS2 copy numbers after four years of treatment. Q-PCR assays have similarly shown no effects of these treatments on the abundances of frequent saprotrophic fungal taxa (e.g., *Pseudogymnoascus*, *Phialocephala* and *Mortierella* spp.) in soil at Kongsfjordneset after 3 years [30]. These observations are in broad agreement with studies showing delayed responses of ectomycorrhizal and saprotrophic fungi to summertime warming in Arctic soils [73,74], possibly owing to the longer turnover times in soil of fungi than bacteria [75,76]. They also corroborate studies showing no effects of warming on soil ergosterol concentrations (an estimate of soil fungal biomass) after 7–15 years of treatment [11,77]. Nevertheless, fungal ITS2 copy numbers exhibited similar, but not statistically significant, responses to the warming treatment to those of 16S gene copy numbers. Assuming that the effects of warming on the size of the soil microbial community recorded here are cumulative in nature, then we anticipate increased fungal DNA concentrations in warmed soils at subsequent samplings of the experiment at Kongsfjordneset. Furthermore, we anticipate that the warming treatment might also alter fungal community diversity and composition, as found in temperate soils with dense woody plant cover, in which 10 years of heating reduces soil fungal diversity and alters community composition [78]. These effects are driven largely by changes to the abundances of ectomycorrhizal *Cortinarius* and *Russula* species [14,78], the former of which are present in the roots of *Salix polaris* on the Brøgger Peninsula [79].

### 4.6. Sparse Effects of Irrigation on Gas Fluxes and the Soil Microbial Community

Given the strong effects of water availability on microbial activity, we anticipated that irrigation would increase CO_2_ efflux and other processes associated with a more active soil microbial community, as it does in annually irrigated soils in the Canadian High Arctic [13] and in soils irrigated weekly during summer in northwestern Greenland [23]. However, other than a marginally significant 15% increase in CO_2_ efflux from irrigated soil at Kongsfjordneset, the irrigation treatment, which had been most recently applied 50–53 d before the measurements reported here were made, had no main effects on the activity, abundance or diversity of the soil microbial community. This is consistent with the lack of effect of irrigation on VSWC recorded here, and observations from summers 2015–2017 similarly showing irrigation to have no effect on gravimetric soil water content *c*. 60–80 d after treatment [30]. Had water been applied to soil more frequently, then significant irrigation effects on VSWC and the soil microbial community may have been recorded. Nevertheless, the frequency of the treatment is broadly representative of precipitation on the Brøgger Peninsula, which is very sparse during summer and tends to be restricted to short, intense periods of rain, with heavy rainfall accounting for approximately one quarter of annual precipitation [4]. Thus, while it remains possible that soil microbial communities on the peninsula will respond to the increased intensity of summertime rainfall events over the next century [4], the observations here suggest that long-term changes to the communities will only be forced by substantial increases in the frequency of rainfall during summer.

### 4.7. Predicted Effects of Warming on Brøgger Peninsula Soils

Long-term studies at Bayelva on the Brøgger Peninsula, 9 km south-east of Kongsfjordneset, have recorded an increase of 3.6 °C in the mean temperature of patterned ground soil at 40 mm depth between 1998 and 2017 [6]. As with air temperature [4], the rate of increase in soil temperature during summer at Bayelva is slower, with a rise of 1.6 °C having been recorded during this season at 40 mm depth in 1998–2017 [6]. Given that the patterned ground soils at Kongsfjordneset and Bayelva have similar moisture properties, with those at the latter location also having summertime VSWCs of *c*. 0.30 m^3^ water m^−3^ soil [6,80], then it seems reasonable to conclude that their temperatures during summer will rise at similar rates as the climate of Svalbard warms. Since soil temperatures on the Brøgger Peninsula are currently rising at 0.8 °C per decade during summer [6], and the effects recorded here were elicited by a 1 °C rise in summertime soil temperature, then we anticipate that similar effects will manifest themselves in the natural environment in a little over a decade, by approximately the mid 2030s.

## 5. Conclusions

After 4 years of treatment, a 1 °C rise in the mean summertime temperature of a sparsely vegetated patterned ground soil on Svalbard increased CO_2_ efflux and CH_4_ consumption rates by 44% and 78%, respectively. As in studies on barren maritime Antarctic soils, warming elicited rapid microbial community responses, supporting the views that interactions with plants in vegetated Arctic soils may be responsible for the decadal lag times of microbial communities to warming treatments, and that bacterial community size in soils with initial C:N ratios of >37 responds positively to warming. At current rates of soil warming during summer on Svalbard, we anticipate that similar effects to those reported here will become apparent in patterned ground soils on the Brøgger Peninsula by the middle of the next decade.

## Figures and Tables

**Figure 1 biology-11-01819-f001:**
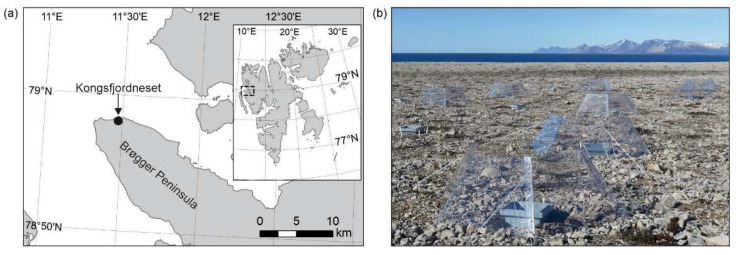
(**a**) Map showing the location of Kongsfjordneset. Inset shows the location of Svalbard, with the dashed box denoting the area shown in the main Figure. (**b**) Image of the field experiment at Kongsfjordneset, showing eight of the 24 open top chambers (OTCs) used to warm soil. The OTCs have a basal diameter of 1.04 m. Note the stainless steel frames used to trap gases during flux measurements.

**Figure 2 biology-11-01819-f002:**
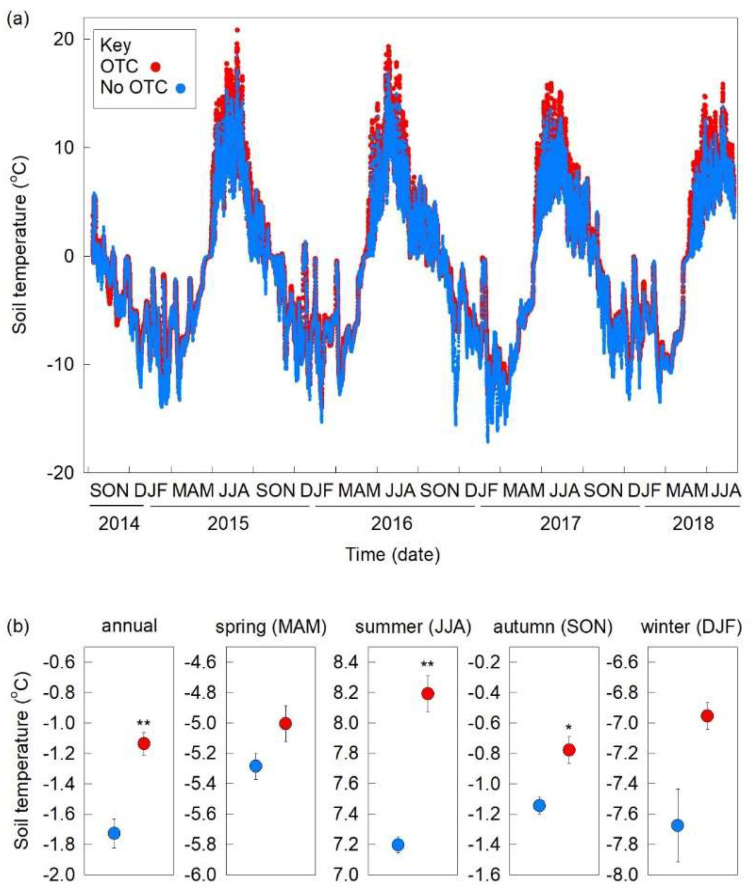
(**a**) Soil temperatures recorded hourly at 30–35 mm depth in unchambered and chambered plots between September 2014 and August 2018. Values are means of four measurements, except those for unchambered plots between September 2017 and August 2018, which are means of two measurements. See key for treatment notation. (**b**) Mean annual and seasonal soil temperatures measured at 30–35 mm depth in unchambered and chambered plots. Values are means of four replicates ± SEM. Differences between each pair of values are denoted by * *p* < 0.05 and ** *p* < 0.01. Treatment notation as in (**a**). Note that *y*-axes are identically scaled.

**Figure 3 biology-11-01819-f003:**
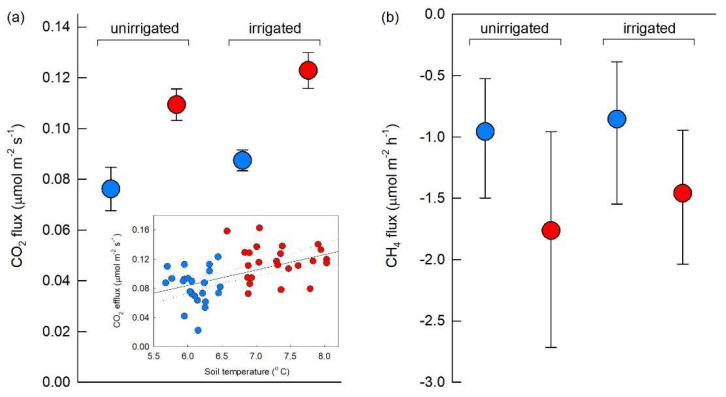
Fluxes to and from soil of (**a**) CO_2_ and (**b**) CH_4_. Note that positive fluxes in (**a**) and negative fluxes in (**b**) indicate the release of CO_2_ from soil and consumption of CH_4_ by soil, respectively. Values are means of 12 replicates, with bars in (**a**,**b**) showing SEM and 95% bootstrap confidence intervals, respectively. Note that *y*-axes are scaled differently. Inset in (**a**) shows soil CO_2_ efflux as a function of soil temperature measured during sampling, with the solid line denoting the best linear fit and the dotted lines denoting 95% confidence intervals. See key in Figure 2a for treatment notation.

**Figure 4 biology-11-01819-f004:**
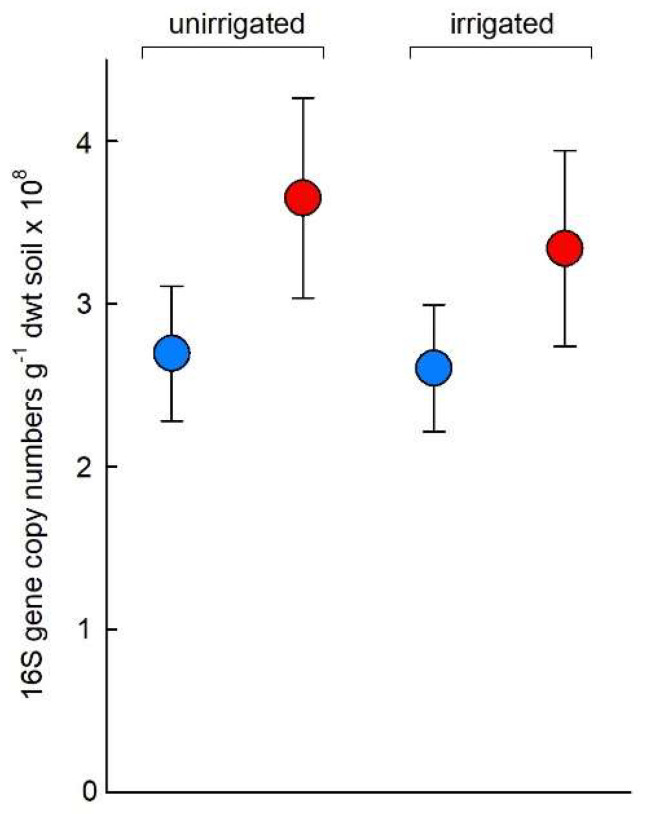
Copy numbers of bacterial 16S ribosomal RNA genes in soil. Values are means of 11–12 replicates ± SEM. See key in Figure 2a for treatment notation.

**Figure 5 biology-11-01819-f005:**
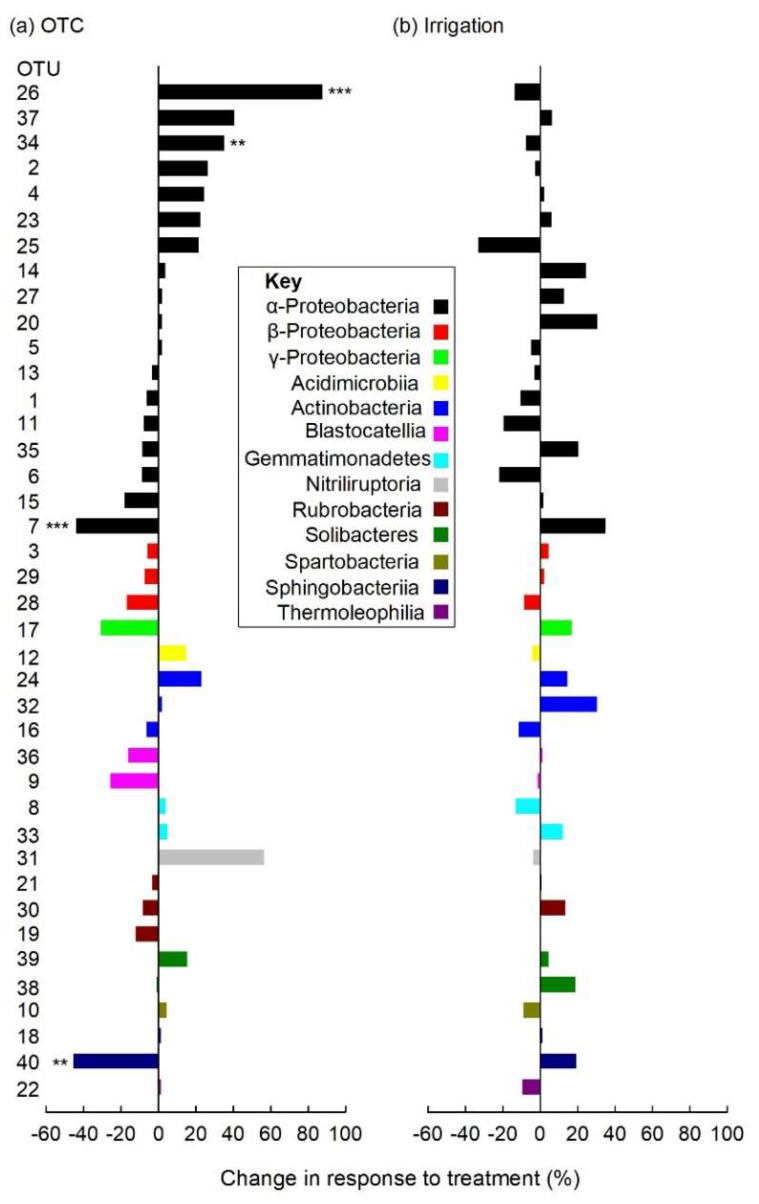
Percentage changes in the relative abundances of the 40 most frequent soil bacterial OTUs in response to the (**a**) OTC and (**b**) irrigation treatments. Values are means of 21–22 replicates. The key shows the classes to which the taxa were assigned at 100% similarity (Appendix A). Note that classes are stacked in the same order in the key as they are in the figure, and that *x*-axes in (**a**,**b**) are scaled identically. Significant effects of OTCs on the relative abundances of four taxa following Benjamini-Hochberg correction are denoted by ** *p* < 0.01 and *** *p* < 0.001.

**Figure 6 biology-11-01819-f006:**
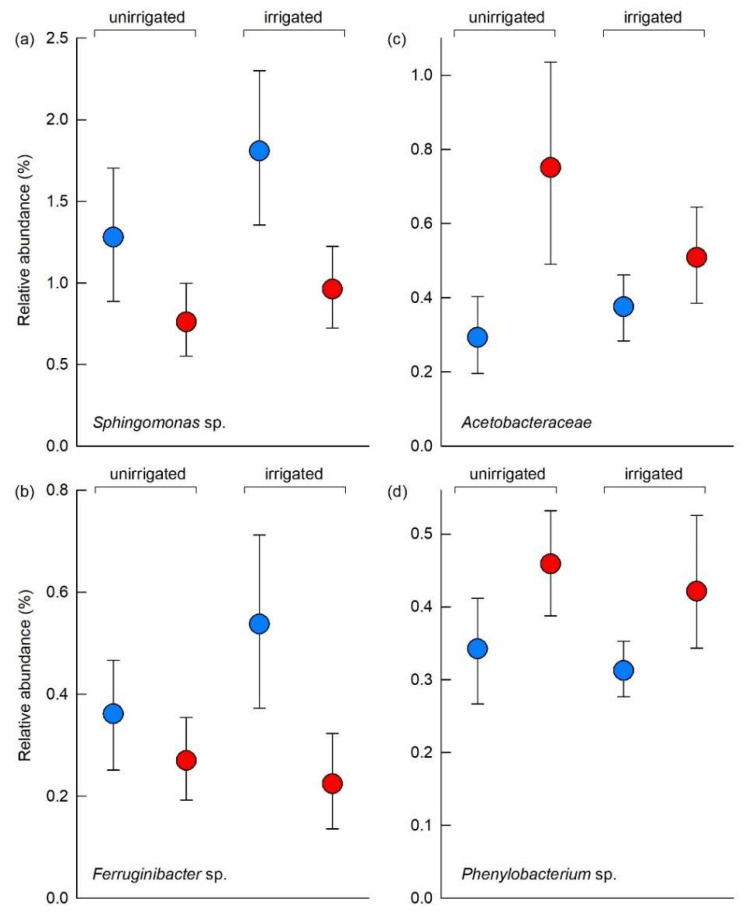
Relative abundances of four soil bacterial taxa that were found to respond significantly to the OTC treatment following Benjamini-Hochberg correction. (**a**) *Sphingomonas* sp. (OTU 7), (**b**) *Ferruginibacter* sp. (OTU 40), (**c**) *Acetobacteraceae* (OTU 26) and (**d**) *Phenylobacterium* sp. (OTU 34). Values are means of 10–11 replicates, with bars showing 95% bootstrap confidence intervals. See key in Figure 2a for treatment notation.

## Data Availability

The data reported here are available at: Newsham, K.K.; Danielsen, B.K.; Biersma, E.M.; Elberling, B.; Hillyard, G.; Kumari, P.; Priemé, A.; Woo, C.; Yamamoto, N. (2022). Greenhouse gas exchange, temperatures, fungal and bacterial abundances and the relative abundances of the 40 most frequent bacterial taxa in a soil warming and irrigation experiment on Svalbard. (Version 1.0) [Data set]. NERC EDS UK Polar Data Centre (https://doi.org/jd4g). The 16S rRNA gene sequence data are available in the NCBI Sequence Read Archive (SRA) BioProject PRJNA888098.

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
