# Peer review of "Rapid Response to Experimental Warming of a Microbial Community Inhabiting High Arctic Patterned Ground Soil"

_biology, 2022, doi:10.3390/biology11121819_

Round 1

Reviewer 1 Report

The article “Rapid response to experimental warming of a microbial community inhabiting High Arctic patterned ground soil” presents previously unpublished and original data. The authors made all the experiments very scientifically and with all the methods used to find out the variable outcome to check the effect of temperature on the soil microbes in the Arctic soils, with molecular diagnosis of soil by ITS and Q-PCR. The overall research, and the findings by the authors are good and appreciated. The research findings could be published in this journal after minor revisions.

Some comments and recommendations are listed below:

Short summary and abstract seem identical so no need to repeat, only abstract is sufficient. 

English grammar and sentence constructions should be extensively improved for the text, as well as check punctuations, spaces, incomplete sentences, spellings, etc. in the MS.

Line 66, ‘abundances of soil microbes is hence fundamental to understanding how Arctic terrestrial ecosystems will alter in the future’ unclear sentence. Rephrase and rewrite.

The unit °C should be without spacing in number and unit for e.g., “1˚C”, and use symbol function for ‘˚C’ throughout the MS.

In figure and table captions, ‘Note’ should be separately mentioned with separate marking as it makes confusion with the figure caption heading. For e.g. *Note: the stainless steel frames used to trap gases during flux measurements, throughout the MS and in supplementary materials as well.

Define the “fwt” and “dwt” and then use abbreviations for the same.

Could it be possible to the authors, to make sub figures in 1, 2, and 3, horizontally for better understanding and spacing?

In table S2: remove commas with points in decimal system or do not use commas.

It is advisable not to use references in the conclusion section. Rewrite this section without references.

References should be checked for spacing, italics, bold, and proper formatting as per the journal guidelines as all the titles of the cited references are italicized so it should be corrected.

Review comments:

 The manuscript could be accepted, after minor revisions.

Author Response

Please note that our responses below are in italics.

The article “Rapid response to experimental warming of a microbial community inhabiting High Arctic patterned ground soil” presents previously unpublished and original data. The authors made all the experiments very scientifically and with all the methods used to find out the variable outcome to check the effect of temperature on the soil microbes in the Arctic soils, with molecular diagnosis of soil by ITS and Q-PCR. The overall research, and the findings by the authors are good and appreciated. The research findings could be published in this journal after minor revisions.

Thank you for these supportive comments.

Some comments and recommendations are listed below:

Short summary and abstract seem identical so no need to repeat, only abstract is sufficient.

The layout of the submitted article was based on a recent paper (Morley et al. 2022; biology11020320) published in the special issue ‘Polar Ecosystems; Response of organisms to changing climate’ and hence included a Simple Summary and an Abstract, as per the journal’s instructions for authors. We checked another article, also published in the special issue (Zenteno-Devaud et al. 2022; biology11121723) and found that it too contained a Simple Summary and an Abstract. We have hence retained both sections in the resubmitted version.

English grammar and sentence constructions should be extensively improved for the text, as well as check punctuations, spaces, incomplete sentences, spellings, etc. in the MS.

The reviewer doesn’t specify where in the article these apparent errors occur, but we suspect that they may have been flagged by the Spelling and Grammar function in Word. We have gone through the text with this function turned on and have altered some, but not all, of the flagged text. The text at L53–55 has been simplified to ‘Mean annual soil temperatures on Svalbard have similarly risen by 3.6 °C between 1998 and 2017 [6].’. That at L322–323 has also been adjusted to ‘The 40 most frequent bacterial taxa responded more strongly to the OTC treatment than to irrigation (Figure 5).’. The long sentence at L333–337 has been split into two, as has that at L369–376. The long sentence at L398–406 has similarly been shortened, and the sentence at L426–431 has been split into two sentences. The long sentences at L491–497, L503–508 and L508–512 have also been shortened. We searched the text thoroughly but failed to find any incomplete sentences or spelling errors.

Line 66, ‘abundances of soil microbes is hence fundamental to understanding how Arctic terrestrial ecosystems will alter in the future’ unclear sentence. Rephrase and rewrite.

To the corresponding author, who wrote the article and whose native language is English, this sentence (‘Determining how warming and increased precipitation will influence the activities and abundances of soil microbes is hence fundamental to understanding how Arctic terrestrial ecosystems will alter in the future.’) scans well and makes sense. However, we have deleted ‘the activities and abundances of’ from the sentence to shorten it. This text has been moved to L82 to clarify the first sentence of the last paragraph of the Introduction.

The unit °C should be without spacing in number and unit for e.g., “1˚C”, and use symbol function for ‘˚C’ throughout the MS.

Changes have not been made to the text because amounts and units, including the degrees symbol, are separated by a space in scientific text (note that Morley et al. (2022) and Zenteno-Devaud et al. (2022) accordingly use spaces before the degrees symbol).

In figure and table captions, ‘Note’ should be separately mentioned with separate marking as it makes confusion with the figure caption heading. For e.g. *Note: the stainless steel frames used to trap gases during flux measurements, throughout the MS and in supplementary materials as well.

The text at L112–113 (‘Note the stainless steel frames used to trap gases during flux measurements.’) scans well and makes sense, and has hence not been adjusted. The other uses of the word ‘note’ (also as a verb) at L254, L278, L281, L317, L531 and L535 similarly scan well and make sense, and changes have hence not been made to the text here, either.

Define the “fwt” and “dwt” and then use abbreviations for the same.

The text ‘fresh weight’ has been used to replace ‘fwt’ at L158. The text ‘dry weight’ has been inserted at L175 so that the text now reads ‘Copy numbers were expressed per g dry weight (dwt) soil’.

Could it be possible to the authors, to make sub figures in 1, 2, and 3, horizontally for better understanding and spacing?

Thanks for this helpful suggestion. Panels (a) and (b) in Figures 1 and 3 have been placed side-by-side to save space. The symbols and text in Figure 2 would have been reduced by about two thirds in size by placing the two panels side-by-side, and so we have retained the previously submitted Figure.

In table S2: remove commas with points in decimal system or do not use commas.

The eight commas in Table S2 have been deleted.

It is advisable not to use references in the conclusion section. Rewrite this section without references.

The references have been deleted from the Conclusions section.

References should be checked for spacing, italics, bold, and proper formatting as per the journal guidelines as all the titles of the cited references are italicized so it should be corrected.

We have checked the References section thoroughly. The only errors that we could find were apparently introduced into the text following submission, with numerous commas in the titles of articles having been converted to semi-colons. These semi-colons have been converted back to commas. In order to conform with the journal’s style, the titles of the cited references, which were unitalicised in the submitted version, remain unitalicised.

Reviewer 2 Report

After reading the manuscript, I accept it in current form.   

Author Response

Many thanks to the reviewer for taking the time to read the manuscript.

Reviewer 3 Report

I have read the manuscript entitled ''Rapid response to experimental warming of a microbial community inhabiting High Arctic patterned ground soil'' (Manuscript ID biology-2026928). This is an interesting study which includes the 4 years field experiment and examines the effect of soil warming on CO2 efflux and CH4 consumption rates as well as the effect on microbial community. The authors have employed adequate and contemporary techniques, presented results in a clear and concise way and anticipated potential environmental effects in the future.

Although the manuscript is well written it requires some changes before publishing. The main remark is that the authors did not perform metabarcoding analyses for the fungal community present in the soil, only for the bacteria. Is it because the warming did not show changes in fungal ITS2 copies in the investigated soil? What about the changes in fungal diversity over the given period? The fact is that fungal community composition can be affected by the soil warming, and although this was a four year study it would be adequate to address this issue and to perform fungal barcoding. See the following reference and discuss the effect of warming on fungi more thoroughly.

Anthony, M. A., Knorr, M., Moore, J. A. M., Simpson, M., & Frey, S. D. (2021). Fungal community and functional responses to soil warming are greater than for soil nitrogen enrichment. Elem Sci Anth9(1), 000059.

Given the abovementioned comments, I suggest a minor revision before acceptance to Biology.

Author Response

Please note that our response below is in italics.

I have read the manuscript entitled ''Rapid response to experimental warming of a microbial community inhabiting High Arctic patterned ground soil'' (Manuscript ID biology-2026928). This is an interesting study which includes the 4 years field experiment and examines the effect of soil warming on CO2 efflux and CH4 consumption rates as well as the effect on microbial community. The authors have employed adequate and contemporary techniques, presented results in a clear and concise way and anticipated potential environmental effects in the future.

Although the manuscript is well written it requires some changes before publishing. The main remark is that the authors did not perform metabarcoding analyses for the fungal community present in the soil, only for the bacteria. Is it because the warming did not show changes in fungal ITS2 copies in the investigated soil? What about the changes in fungal diversity over the given period? The fact is that fungal community composition can be affected by the soil warming, and although this was a four year study it would be adequate to address this issue and to perform fungal barcoding. See the following reference and discuss the effect of warming on fungi more thoroughly.

Anthony, M. A., Knorr, M., Moore, J. A. M., Simpson, M., & Frey, S. D. (2021). Fungal community and functional responses to soil warming are greater than for soil nitrogen enrichment. Elem Sci Anth, 9(1), 000059.

10.1525/elementa.2021.000059

Given the abovementioned comments, I suggest a minor revision before acceptance to Biology.

Thanks for these thoughtful comments. The possible effects of warming on the diversity and composition of the soil fungal community at Kongsfjordneset are discussed at L469–474 (‘Furthermore, we anticipate that the warming treatment might also alter fungal community diversity and composition, as found in temperate soils with dense woody plant cover, in which 10 yr of heating reduces soil fungal diversity and alters community composition [78]. These effects are driven largely by changes to the abundances of ectomycorrhizal Cortinarius and Russula species [14,78], the former of which are present in the roots of Salix polaris on the Brøgger Peninsula [79].’). The reference by Anthony et al. (2021) has been included at L736–737, and an article by Fujiyoshi et al. (2011) has been added at L738–740.